# IoT Security and Computation Management on a Multi-Robot System for Rescue Operations Based on a Cloud Framework

**DOI:** 10.3390/s22155569

**Published:** 2022-07-26

**Authors:** Swarnabha Roy, Tony Vo, Steven Hernandez, Austin Lehrmann, Asad Ali, Stavros Kalafatis

**Affiliations:** Department of Electrical and Computer Engineering, Texas A&M University, College Station, TX 77843, USA; swarnabha7@tamu.edu (S.R.); tvo8275@tamu.edu (T.V.); steven10161999@tamu.edu (S.H.); austin.lehrmann@tamu.edu (A.L.); asadzali89@tamu.edu (A.A.)

**Keywords:** blockchain, image detection, robotics, Robot Operating System (ROS), multi-robot coordination, IoT security, cloud robotics, Unmanned Aerial Vehicle (UAV), Unmanned Ground Vehicle (UGV)

## Abstract

There is a growing body of literature that recognizes the importance of Multi-Robot coordination and Modular Robotics. This work evaluates the secure coordination of an Unmanned Aerial Vehicle (UAV) via a drone simulation in Unity and an Unmanned Ground Vehicle (UGV) as a rover. Each robot is equipped with sensors to gather information to send to a cloud server where all computations are performed. Each vehicle is registered by blockchain ledger-based network security. In addition to these, relevant information and alerts are displayed on a website for the users. The usage of UAV–UGV cooperation allows for autonomous surveillance due to the high vantage field of view. Furthermore, the usage of cloud computation lowers the cost of microcontrollers by reducing their complexity. Lastly, blockchain technology mitigates the security issues related to adversarial or malicious robotic nodes connecting to the cluster and not agreeing to privacy rules and norms.

## 1. Introduction

Beginning in the 1980s with the boom of the Internet, researchers have been motivated to design and build robots that can work together. Research has thus far been mostly focused on improving the coordination and communication, applications, and architecture of multi-robot systems (MRS) [1,2,3]. Unlike humans, a team of robots can perform a task that requires a high level of perseverance and focus. Using multiple sensors and cameras, a multi-robot system can bring robustness to the system through information redundancy, which is essential in search and rescue operations [4]. The present work is focused on building a novel multi-robot coordination system that is secured by blockchain technology, uses drones for searching, and uses rovers to perform rescue operations.

In the past decade, the number of lives lost due to natural disasters ranges from 14,389 in 2015 to as high as 314,503 in 2020 [5]. According to [6], the average cost of a search and rescue (SAR) operation is USD 1375 per person. Multi-robot systems can provide numerous benefits for disaster response. They can often fit into places humans are unable to reach, operate in environments not habitable to humans, operate continuously without sleep, can outperform humans in certain tasks, and most importantly, they are replaceable. Robots can be sent to places no human can access, which is often necessary during disasters. The use of drones and rovers can allow for smaller-scale search and rescue operations that do not require expensive equipment (e.g., helicopters) [7]. Drones provide a field of view from a high vantage point, where telemetry data can provide an accurate description of the location of both the drone and rover. When considering large areas, it requires a large roster of individuals to perform tasks such as delivery or search. This system allows the use of a drone to identify a designated goal from a much higher vantage point and then maneuver the rover towards the goal. This allows for a single person to watch over a large area with fewer personnel.

Recent developments in collaborative robots have heightened the need for improved security in adversarial environments. Current research, including AutoSOS [8] and Centauro [9], has mostly focused on locomotion and coordination and placed less importance on the security of the system. A brief comparison of the ongoing research is shown in Table 1. According to [10], due to the decentralized nature of multi-robot systems the individual robots or nodes communicate via an unprotected communication channel, which makes the system vulnerable to adversarial attacks. Such risks could range from security breaches to loss of confidential data [11]. With the multi-robot system being autonomous, the robots mostly rely on sensors to perceive their environment. With direct knowledge of the sensors, an adversary can tamper with the perceived environment and gain control [12]. SkyJack [13] is a drone hijacking software that can change the SSID of a drone and establish a malicious connection. This results in Byzantine faults in the MRS, which are difficult to identify [14,15]. Blockchain technology [16] has been highly successful in achieving Byzantine Fault Tolerance due to its consensus algorithm [17]. Studies over the past decade have used the concepts of blockchain in finance sectors and expanded to AI and robotics [18,19]. In [20], Strobel et al. compared the results of classical and blockchain approaches on a collective decision-making scenario in both the presence and absence of Byzantine robots using the ARGoS [21] swarm robotics simulator. Ahmad et al. [22] carried out a comprehensive survey on cloud attacks, security issues, present challenges, and limitations in cloud computing, and highlighted the importance of securing cloud logs through blockchains.

A blockachain is a type of shared database or ledger that is maintained by all the members within a network. It differs from a typical database in that a blockchain stores data in blocks that are then connected via cryptography. This work is an implementation of blockchain technology in a multi-robot system. When a new rover is connected to the network, a new block is added to the blockchain, which stores the hash of the previous rover and maintains the database of all connected rovers within the network. Computations and communications are offloaded to the cloud, and a website was built that can track the progress of the surveillance, as shown in Figure 1. All rovers or nodes connected to the system were built using Robot Operating System (ROS) [27] and are a part of the blockchain private to that network. This is a novel work that uses the security of blockchain technology to secure a multi-robot communication system built on a Cloud framework.

Additionally, in the recent literature efficient networks such as LNSNet [28] and ESPNet [29] were created with the purpose of allowing for mobile usage by reducing power usage and size of the model. In [25], UAV and UGV collaboration were highlighted in an indoor environment using semantic segmentation from a blimp and Simultaneous Localization and Mapping (SLAM) for local mapping and obstacle avoidance. A priority in this analysis was real-time computations done on the robots, so LNSNet was used for semantic segmentation. This work offloads these computations onto the cloud, which is a novelty in UAV-UGV collaboration, by removing limitations present in mobile applications. A Fully Convolutional Network (FCN) [30] with ResNet-50 [31] backbone was used.

## 2. Methodology

This work is scoped to successfully find a specified item in a small plot of land. On the website, the user can see all rover sensor readings and camera footage. Most heavy computations (i.e., machine learning inference and global path planning) are done on the cloud. Figure 2 shows the overall system, including parts that are separate from the cloud.

When provided with an area to search and an item to find, the drone provided vision which was sent to the cloud. After processing, a set of coordinates are sent to the rover and the movement is executed. The user had a full view of this situation from the web page, where the user can look at camera footage and sensory data.

### 2.1. Blockchain for Distributed Authentication

In certain scenarios, it is very important that untrusted third-party actors are not able to read communications between network members (i.e., vehicles). The classic way to deal with this is through encryption of the data [32]. Most of the top commercial cloud and IoT services have created their own encryption techniques, as shown in Table 2. Traditional encryption techniques are more vulnerable to manipulation by a malicious administrator or outside hackers because they lack immutability. Blockchain uses Proof of Work to validate a transaction [33], which provides resiliency, integrity, and privacy and builds trust among the agents. This work presents a simple private blockchain network that was built on top of the IoT system. Each node in the network had a public key and a private key generated with the RSA algorithm [34]. Every node stored every other node’s public keys, and each message was signed with the public key of the sender. A node only accepted a message if it was signed by a private key that matched a public key it had on file [See Figure 3].

To start with, each vehicle and the controlling cloud computing node had an RSA-generated 512-bit private key and public key. Every network member had all the public keys, but only the node itself had access to its private key. Each member also had a copy of the blockchain [41]. As shown in Figure 3, each block had data, its own hash, and the hash of the block before it. Peer-to-peer verification is achievable because each node possesses a copy of the blockchain. In the event of a tie, the chain with the most blocks is picked. In this implementation, the data that reside on the blockchain are the public keys of member variables. Whenever a node sends a message (in JSON format) to all the other nodes, a signature field is required, which is where the node uses its private key to sign the data. Each public key on the blockchain of the node that receives this transmission is used to attempt to verify the signature in a linear fashion. Only the public key connected to the private key has the ability to verify it. Therefore, if one of the blockchain’s keys can confirm it, the message was from a real member. If not, the message is not to be trusted.

In this implementation, the network had six opcodes, which were the operations the network members could perform. For a message to be legitimate, it must have an opcode field, a data field, and a signature field. The six opcodes are as follows: ADD_BLOCK, INTRUSION_ATTEMPT, SENSOR_DATA, BROADCAST_BLOCK, ERROR, and CONFIRMATION. They are described in detail in Table 3.

### 2.2. Rover Operating via ROS

In this work, the Husarion ROSbot 2.0 [42] rover has been used which functions under ROS. It is equipped with LiDAR and IMU sensors which are responsible for local path planning and communication metrics. ROS was used to control the movements and generation of status metrics, while a Python 3 script with Boto3 was used for polling of the coordinates and publication of the IMU metrics to AWS.

After the cloud-generated coordinated were downloaded, the navigation stack as shown in Figure 4 executes them in order. The data from the LiDAR sensor are sent to slam_gmapping to create the local map and move_base is executed.

The move_base package provides an implementation of an action that, once given a goal in the world, attempts to reach it with a mobile base. The move_base node links together a global and local planner to accomplish its global navigation task. The move_base node also maintains two cost maps: one for the global planner, and one for a local planner used to accomplish navigation tasks. The /drive_controller node subscribes to the /pose topic, which contains the rover’s orientation and position. It uses this information and publishes a transform.

### 2.3. Cloud Computations

The cloud computations are responsible for receiving and processing sensory data to create movement instructions for the rover. These movement instructions include a set of coordinates that start at the rover and end at the objective.

The cloud service platform used was Amazon Web Services (AWS), of which SageMaker and Lambda were used for processing and a Simple Storage Service (S3) was used for storage. SageMaker is a cloud machine learning platform, and was used to label images, train models, and deploy them for inference. Lambda is a serverless computing platform, and was used to execute scripts for processing.

For the creation of the machine learning model, Unity was used to simulate a drone’s footage, where the scene contained the rover, obstacles, and objectives. The data were created by randomizing each object’s position and orientation. As the dataset was simulated, there were fewer random elements present when compared to real drone footage. The benefit is that there was less time required for the creation of the model.

The machine learning task used was semantic segmentation, where the choice of algorithm, backbone, and optimizer were based on accuracy over a short period of training time. An FCN algorithm, ResNet-50 backbone, and RMSProp [43] optimizer provided the best accuracy. The learning rate was tuned using Bayesian optimization. The decay and momentum factor were used in RMSProp, and these parameters were left at their default values. The hyperparameters for the best performing model are shown in Table 4.
(1)$$E\left[g^{2}\right]\left(t\right)=\gamma E\left[g^{2}\right]\left(t-1\right)+\left(1-\gamma \right){\left(\frac{\partial c}{\partial w}\right)}^{2}$$
(2)$$w\left(t\right)=\alpha w\left(t-1\right)-\frac{\eta }{\sqrt{E\left[g^{2}\right]\left(t\right)}}\frac{\partial c}{\partial w}$$

Equation (Equation 1) is the moving average of squared gradients used in RMSProp. γ is the decay factor, and ∂c/∂w is the gradient. Equation (Equation 2) is the weight equation with a momentum factor α.

The pipeline used for the cloud computations can be visualized in Figure 5. The first AWS Lambda function was to run inference, and this was implemented using a SageMaker SDK layer to call the deployed model for inference. The function first obtains an image from S3, then runs inference to create the highest probability mask image. The resulting mask is then stored back into S3 for the second AWS Lambda function.

The second AWS Lambda function was to run the global path planning, convert coordinates systems, and send them to the rover. This procedure can be visualized in Figure 6. To process the mask, an OpenCV layer was utilized. The path planning algorithm used was A* with a greedy heuristic [44]. Specifically, the Euclidean distance was used to calculate the distance from the current search node and the goal node, as shown in Equation (Equation 3); the pseudocode is shown in Algorithm 1. Greedy heuristics have been used in past works when considering global path planning with UAVs [7].
(3)$$\mathrm{Euclidean}\;\mathrm{Distance}=h\left(n\right)=\sqrt{{\left(x_{2}-x_{1}\right)}^{2}+{\left(y_{2}-y_{1}\right)}^{2}}$$

After coordinates were created from the mask, they were converted to the rover’s coordinate system by changing the domain and range of the grid to have the origin centered with positive and negative values. All coordinates were translated to force the rover to be at the origin of the grid.
**Algorithm 1:** A* shortest path Algorithm with Greedy heuristic**Require:** 
maze, start and goal**Ensure:** 
start≠goal1:**procedure**A_Star(maze,start,goal)                                                                                                   ▷ returns path2:    Add start to priorityQueue3:    searchedNodes←ϕ4: 5:    **while** priorityQueue≠ϕ **do**6:        **if** currentNode←goal **then**7:           return path8:        **end if**9:        Remove currentNode from priorityQueue10:      Add currentNode to searchedNodes11: 12:      child←neighbor.currentNode13: 14:      **for** each child in neighbor **do**15:           **if** child is in searchedNodes **then**16:               continue17:           **end if**18:           child.g←currentNode.g+d    ▷*d* is distance between currentNode and child, which is 1 in this case19: 20:           child.h← distance from child to goal21: 22:           child.f←child.g+child.h       ▷*h* is calculated using heuristic function, Euclidean distance in this case23: 24:           **if** child.position is in priorityQueue **then**25:               **if** child.g>priorityqueue.child.position.g **then**26:                   continue27:               **end if**28:           **end if**29: 30:           Add child to priorityQueue31:        **end for**32:    **end while**33: 34:    return failure                              ▷ If priorityQueue is ϕ35:**end procedure**

### 2.4. Website for Monitoring

The website included features that monitored the stats of the drone and the rover. The website acted as the central hub for all monitoring to ensure that the drone and the rover were performing correctly. The front end was built using a ReactJS framework, and all routing was done using React Router Dom. AWS Amplify was used to host the website, and AWS Cognito was used to support the Login system.

## 3. Results

For the cloud computations, machine learning and global path planning were peformed and shown to effectively offload computation to the cloud. For the rover, it managed to reach its objective coordinates based on the local mapping created from its LiDAR sensor. For the system, the coordinates were obtained by the rover and its protocol was successfully executed. This protocol included reaching the objective and uploading its status metrics to display on the website.

### 3.1. Semantic Segmentation Model

Using the Unity simulation, 500 images were created for training; the training and validation data set were split into an 80:20 ratio (400 training and 100 validation). The GPU instance used for training was ‘ml.p2.xlarge’ using an NVIDIA Tesla K80 Accelerator, 4 vCPUs, and 61 GB of memory [45]. Three main metrics were considered: training loss, pixel accuracy, and mean intersection over union (mIoU). Pixel accuracy measures the correct prediction of each pixel over all pixels, while the intersection of union measures the overlap of masks for each class. The best performing model’s tuning hyperparameters are shown in Table 4, and the graph of the metrics are shown in Figure 7, Figure 8 and Figure 9.

When tuning for maximum mIoU, the maximum number of epoch was set to 50. The best performing model ran for 39 epochs, at which point early stopping was initiated due to the lack of substantial improvements in the model. From Figure 7, the training loss is shown to gradually decrease then level off, which indicates a healthy model. For the two performance metrics shown in Figure 8 and Figure 9 the curves look similar, with a gradual increase and eventual leveling. However, it is important to note that the pixel accuracy score ranges from approximately 0.96 to 0.99, while the mIoU provides a more realistic score range of 0.5 to 0.85. The reasoning behind this biased score is that the images provided had mostly background elements, as shown in Figure 10b.

When running inference on AWS Lambda, the instance used was ‘ml.g4dn.xlarge’ using an NVIDIA T4 GPU, 4 vCPUs, and 16 GB of memory [46]. Using this instance resulted in an average inference time of 2.5 s for each image. To show that there was an offload of computation, images were batch processed for approximately 6 min and CPU and GPU utilization were measured. From Figure 11, it can be seen that the CPU and GPU utilization were increased by approximately 30% and 15% respectively. Table 5 compares different parameters from this work to previous works that used efficient networks for mobile application.

### 3.2. The Path Planning Algorithm

After the mask was created, a path was generated from it using the A* algorithm. In order to lower computation time, the mask was down-sampled by a factor of 10, effectively reducing the search space to 1% of the original mask. As the Unity frames were randomly generated, most images were similar to the one shown in Figure 10a, with no obstacles between the start and goal. The average computation time for these cases was very fast, at approximately 0.5 s.

### 3.3. Rover Obstacle Avoidance

The rover navigated to each coordinate provided by the cloud computations. This was done by incorporating a local occupancy grid-based path planner. The rover took in the LiDAR data and plotted a path to reach the coordinate while avoiding any obstacles lying between. In order to validate this system, a test bench was created called “Titanic”, shown in Figure 12. The “Titanic” test bench had an obstacle in between the rover and the coordinate (visualized using a white post-it note). The goal was to reach the coordinate without colliding with the obstacle and return to where it started while avoiding the obstacle again.

When tuning the navigation stack, the “Default” was defined to be the first iteration of the stack that resulted in three of five successful attempts. Additionally, the greatest factor in reducing the time to and from the coordinate was the movement capabilities of the rover. The ability to make tighter turns, in particular, allowed for reduced collisions and less time spent adjusting.

The greatest factor in the reliability of success was the map update frequency. This allowed the rover to make adjustments faster and prevent collisions when moving at high rates of speed. Figure 13 displays the performance with different tuning parameters.

### 3.4. Rover Communication

The rover stayed stationary until the coordinates were polled from the cloud (AWS). This was done using the Boto3 library and checking whether a text file in S3 has changed file sizes, which indicated new coordinates. After the coordinates are pulled from S3, it copied the coordinates and uploaded a blank text file to replace the one in S3. This served to prevent the next poll to execute old coordinates.

When the rover began to move, it sent its metrics from the IMU to the cloud to display on the website. This is done in a similar way as uploading the metrics to S3, and the website polled S3 to update itself. Figure 14 demonstrates this process.

### 3.5. System

The whole system starts initiated with a wrapper function that executed the the inference and created the the rover coordinates. The rover coordinates are written into an empty text file located in S3. The rover polls S3 for changes in this text file, and when a change is noticed it executed its protocol. Blockchain implementation means that only trusted messages are read.

Two cases were tested: one without obstacles between the start and goal and one with an obstacle. The case with no obstacle was used to test whether the complete system was able to function with networking, while the case with an obstacle was otherwise the same while testing the obstacle avoidance on the rover. The case with no obstacle shown in Figure 15, while the one with an obstacle is shown in Figure 16.

The timing performance was split between processing time on the cloud and network latency. The average measured processing time on the cloud was 2.5 s for inference and 0.5 s for path planning, which sums to 3 s of processing. The AWS region used was us-east-2 (Ohio); thus, the latency was dependent on the location of the network used by the rover. The average latency when testing ranged from 50 ms to 100 ms.

## 4. Discussion

In the present day, with improvements in 5G, dependable internet, intelligent mobile devices, IoT infrastructures, and AI-based business intelligence platforms, there is a need to review how traditional cloud services are managed and solve related security risks. To strengthen data storing methods in the cloud and data security, blockchain technology with decentralized cloud storage is useful, and this strategy protects the stored data from alteration and deletion. The design proposed here uses blockchain technology to secure linked devices, making cloud systems impenetrable and boosting user confidence in the cloud environment.

In recent works involving UAV-UGV SAR processing has been carried out onboard, and the computation requires dedicated hardware. Hardware can be expensive, and costs scale with the number of robots used. In this work, the cloud was introduced to reduce onboard processing. This can lead to reduced upfront hardware costs and allow for increased flexibility and scalability. As most heavy computations are performed on the cloud, dedicated hardware can be removed from the robots, lowering the total cost. Furthermore, more robots can be deployed simultaneously which can effectively lower search time. Lastly, updates and changes to the computations can be carried our remotely.

Recent studies that approach real-time computer vision tasks require specialized networks, such as LNSNet and ESPNet, to create fast and efficient models. While these models are smaller in size and require less powerful GPUs, they have lower accuracy when compared to models created from large networks. In our work, all processing and models are located and carried out in the cloud, and thus limitations of hardware and size are not present. GPU instances are used to run inference, and they can be swapped to increase timing performance. Furthermore, model sizes are redundant in this work, as shown by the model size in this work being 256 MB, while LNSNet and ESPNet were both less than 2 MB.

The implementation of rover-based local pathfinding and obstacle avoidance allows for fewer generated coordinates. This translates to less data having to be sent and stored. Additionally, the local obstacle avoidance allows for path adjustments in real time for use in more dynamic environments. Using a cloud-based global path planner allows for large scale path generation. This, paired with the local path planner on the rover, provides a quickly reacting and more knowledgeable navigator compared to a single robot solution. While we have not tested how weak network connectivity affects the blockchain approach, we believe that it will not affect the performance of the entire system, as all noisy communication channels with be verified in a peer-to-peer manner. Any mismatch due to loss of information will be considered invalid.

## 5. Conclusions and Future Work

This work serves as a proof of concept of a cloud-based UAV-UGV and blockchain secured approach to search and rescue. The drone detects an object and sends the relevant information to the cloud for computing. The processed information is sent back to the rover for it to move to the object’s location. This communication is secured using blockchain. After receiving coordinates from the cloud, the rover navigates to the object while avoiding obstacles using its LiDAR sensor. The drone provides vision to the cloud where it creates a map of the area that is being considered. This allows for optimal path planning, as it provides a second viewpoint that shows the “full picture”. Without a drone, a local path planning robot needs to make sub-optimal decisions, resulting in lost time.

For future work, usage of real drones will be considered. Footage from a drone can add randomness to the machine learning model and to testing the overall system. As a machine learning model will be more complicated than the one created from the Unity simulation, different tasks will be considered, e.g., object detection. Additionally, different path planning algorithms can be tested with fewer dependencies from local mapping. In [7], three different algorithms were considered: greedy heuristics, potential-based heuristics, and partially observable Markov decision process-based heuristics. Using different algorithms can reduce the need to rely on the rover’s local path planning, which reduces the required computations on the robots. Currently, only a drone and a rover have been used to as a part of the blockchain, and the blockchain used was very small. More rovers can be introduced to the network to compare the delay times in verifying the authenticity of an external rover as the blockchain grows longer.

## Figures and Tables

**Figure 1 sensors-22-05569-f001:**
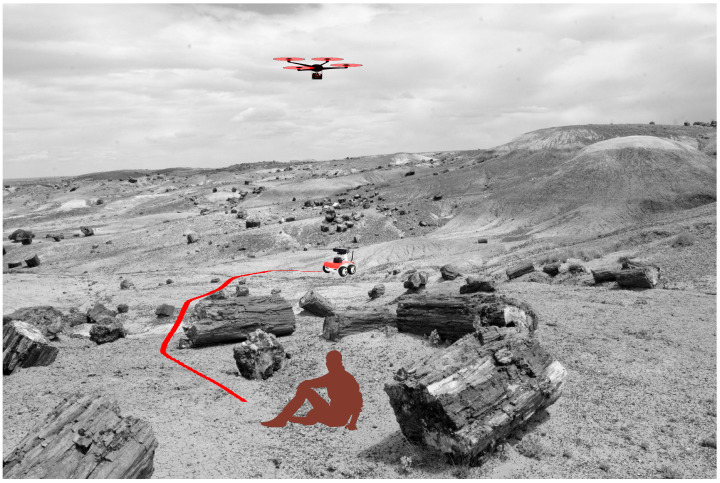
Conceptional illustration of a search and rescue mission utilizing a drone and rover.

**Figure 2 sensors-22-05569-f002:**
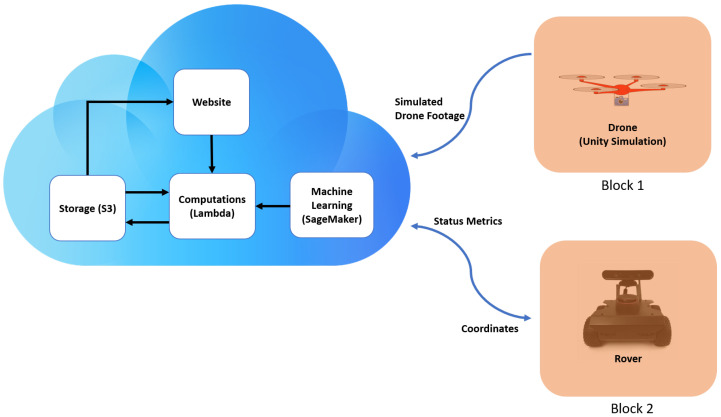
System diagram highlighting parts involved with Cloud (AWS). Block1 and Block2 are the vehicles connected to the network and serve as individual blocks of the blockchain.

**Figure 3 sensors-22-05569-f003:**
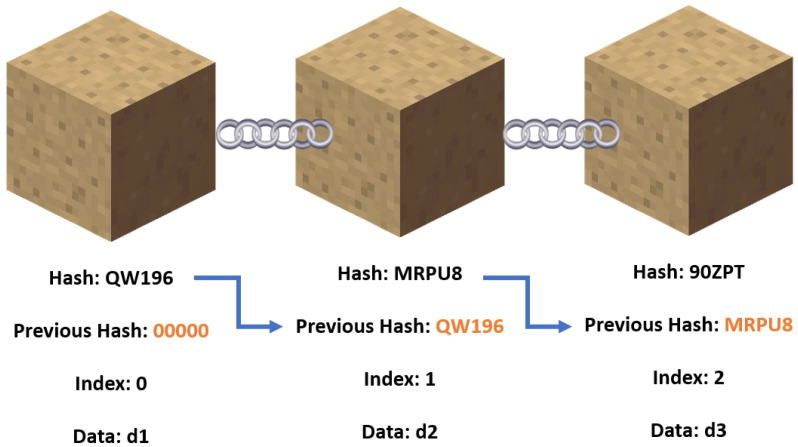
Pictorial representation of the Blockchain: To maintain the chain integrity, the preceding block’s hash must be located in the current block.

**Figure 4 sensors-22-05569-f004:**
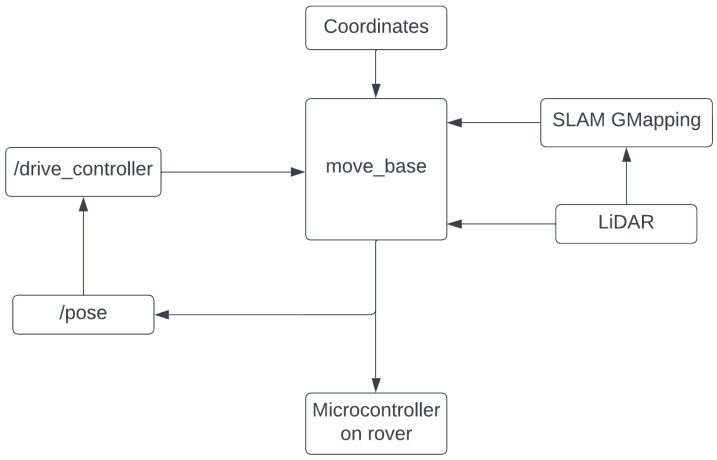
Navigation stack of the rover.

**Figure 5 sensors-22-05569-f005:**
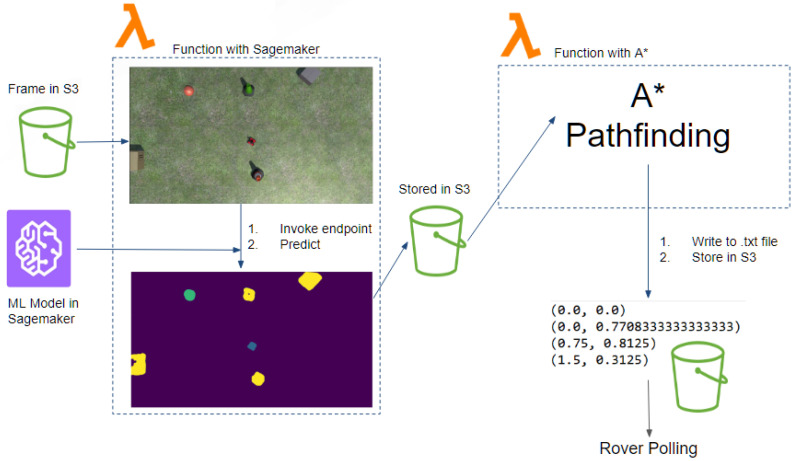
AWS pipeline for processing and sending coordinates to the rover.

**Figure 6 sensors-22-05569-f006:**
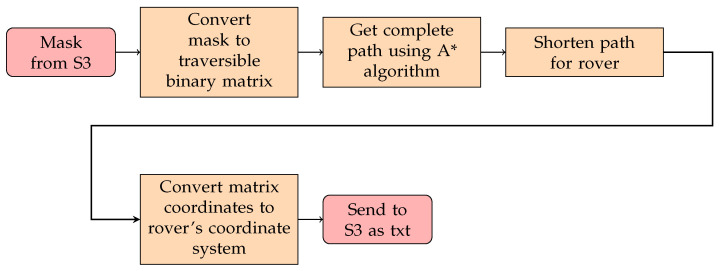
Procedure to obtain rover coordinates.

**Figure 7 sensors-22-05569-f007:**
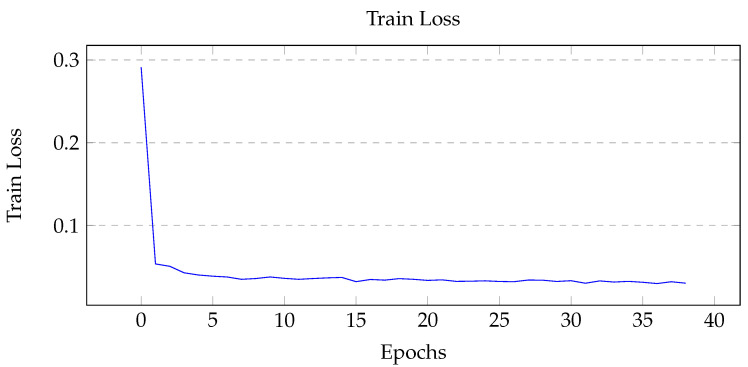
Graph of training loss over each epoch. Initial large decrease for the first ten epochs followed by a gradual decrease.

**Figure 8 sensors-22-05569-f008:**
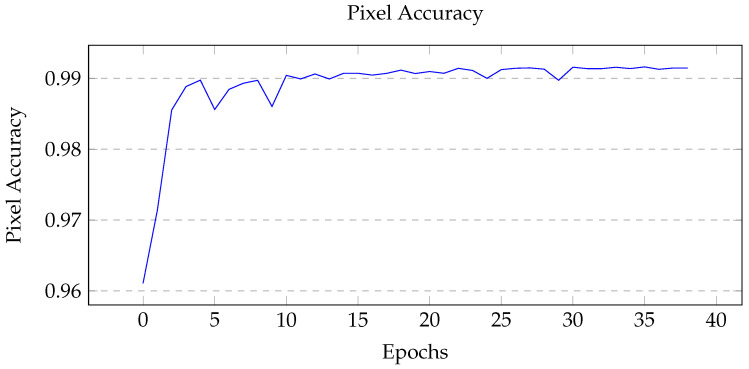
Graph of pixel accuracy over each epoch. Initial increase then gradual improvements. Notice the range of pixel accuracy.

**Figure 9 sensors-22-05569-f009:**
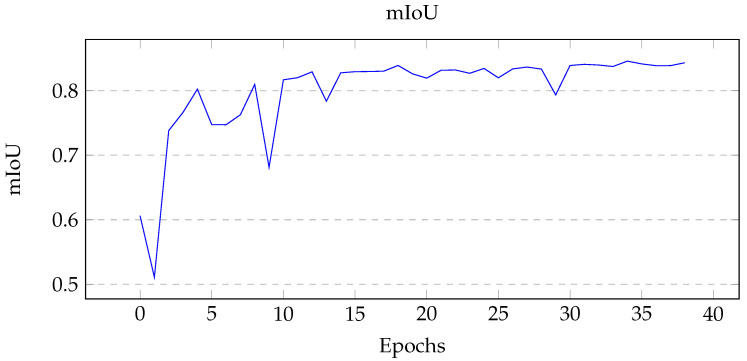
Graph of mean intersection over union (mIoU) over each epoch. A similar graph to the pixel accuracy, although notice the more realistic range.

**Figure 10 sensors-22-05569-f010:**
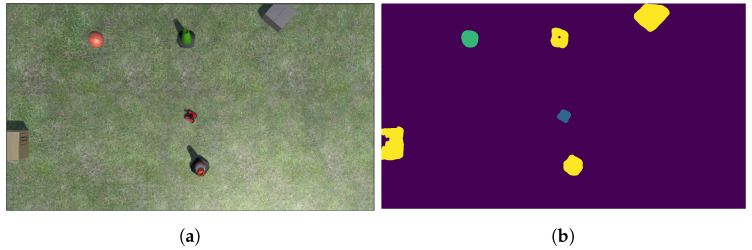
(**a**) An example Unity image. (**b**) The resulting mask from the best performing model.

**Figure 11 sensors-22-05569-f011:**
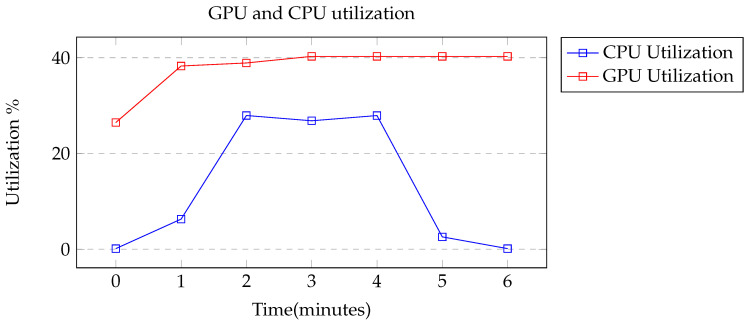
Graph of CPU and GPU utilization over 6 min of batch inference.

**Figure 12 sensors-22-05569-f012:**
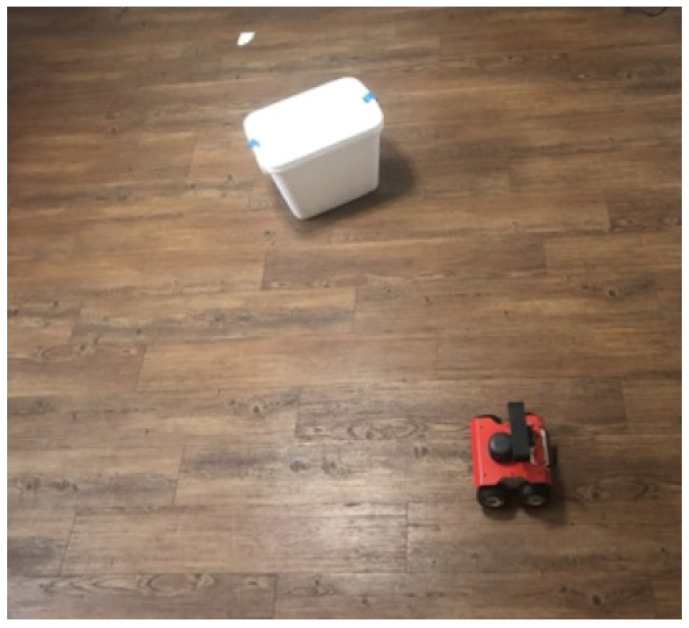
The “Titanic” test bench.

**Figure 13 sensors-22-05569-f013:**
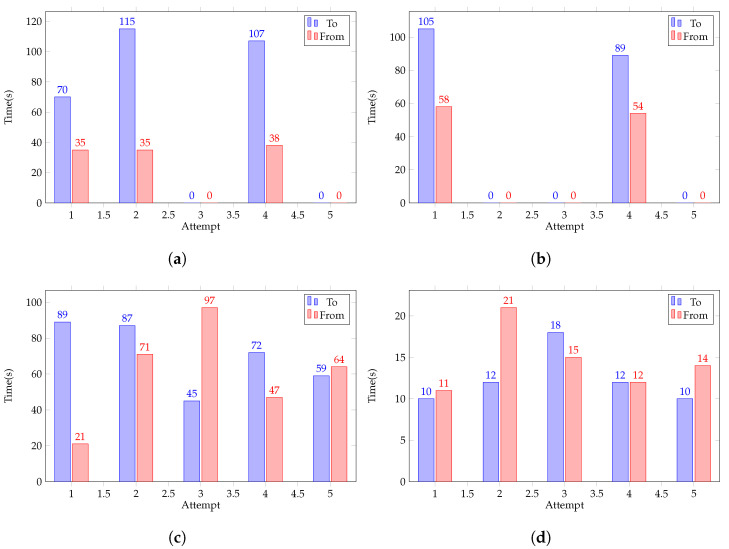
(**a**) Default test bench results. (**b**) Test bench results with only forward speed increase. (**c**) Test bench results with both forward speed and update frequency increase. (**d**) Final tune test bench results.

**Figure 14 sensors-22-05569-f014:**
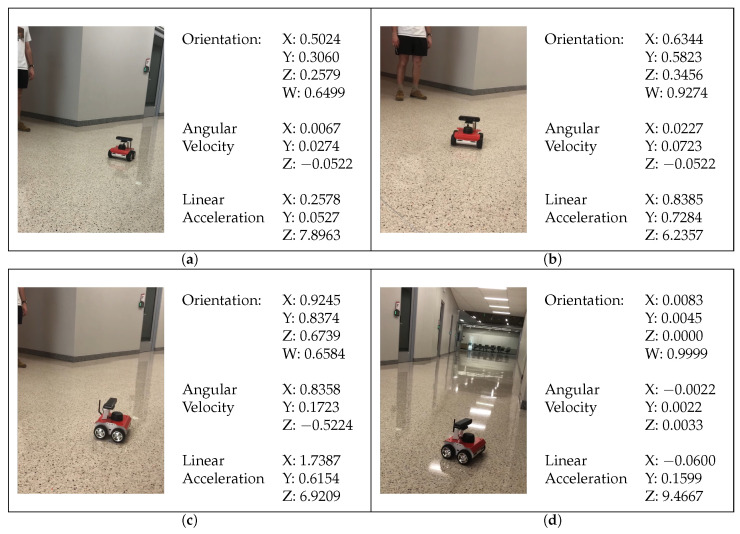
(**a**) Initial rover position status metrics from the website alongside real world reference. (**b**) Second rover position. (**c**) Third rover position. (**d**) Final rover position. Notice the orientation was reset to near zero; this indicates that the rover has reached the goal.

**Figure 15 sensors-22-05569-f015:**
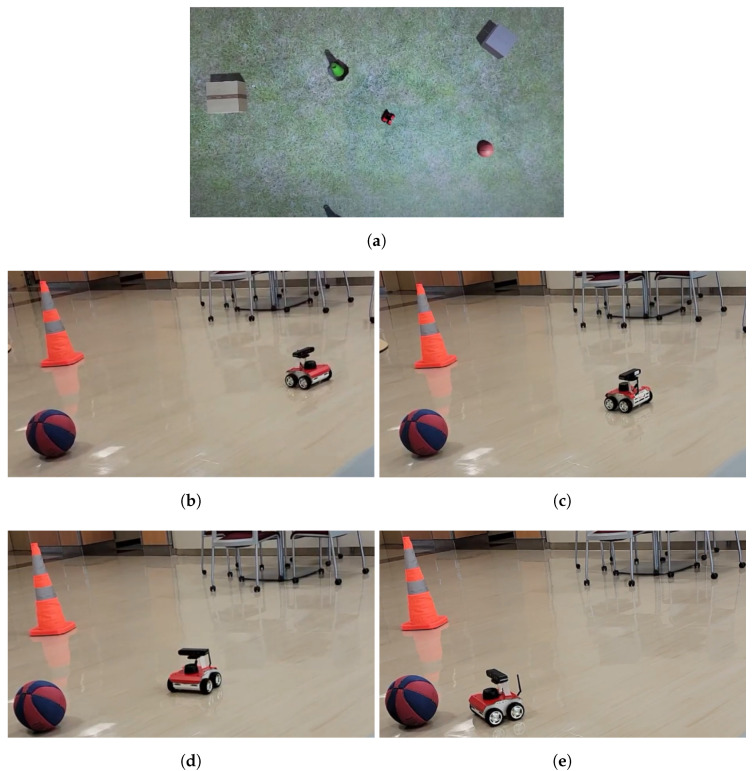
(**a**) No obstacle Unity frame. (**b**) The initial position was manually setup. (**c**) Second coordinate position. (**d**) Third coordinate position. (**e**) Final coordinate position, where the rover has reached the goal.

**Figure 16 sensors-22-05569-f016:**
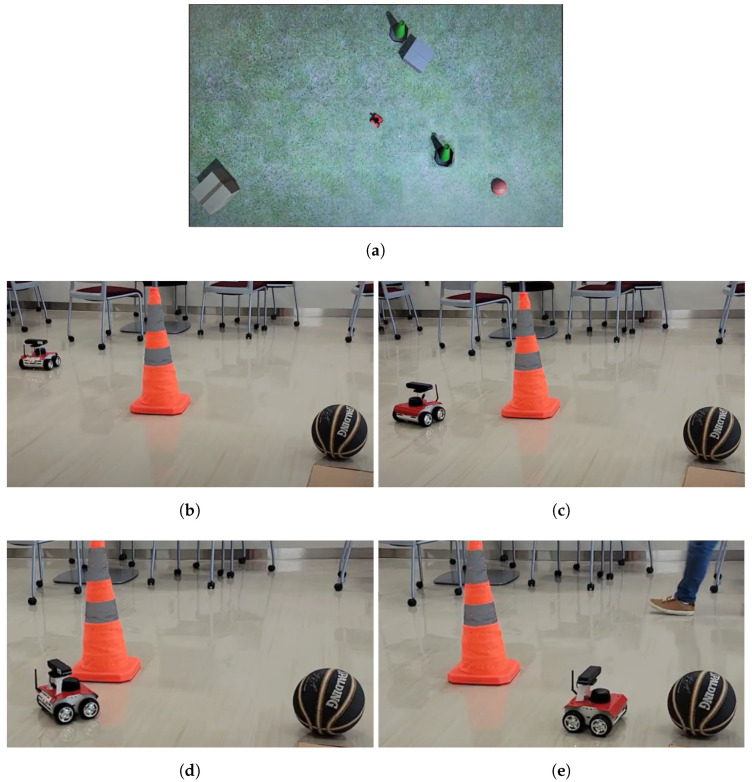
(**a**) One obstacle Unity frame. (**b**) Initial position. (**c**) Second coordinate position, where the rover’s obstacle avoidance keeps a distance from the cone. (**d**) Third coordinate position, where the rover swerved around the cone. (**e**) Final coordinate position, where the rover has reached the goal.

**Table 1 sensors-22-05569-t001:** This is a comparison of work carried our in Search and Rescue (SAR) Robotics. We compare the existing literature based on whether the system is autonomous or not, where the data are processed, and whether a cloud is involved. We compare the architecture of the system, i.e., whether it is distributed or centralized. The scenario categorizes what kind of SAR environment the work focused on. The security protocols used in different works are compared as well.

	Description	Auton.	Sensory Processing	Distribution Control	Scenario	Security	Cloud
AutoSOS [8]	Multi-UAV supporting maritime SAR with lightweight AI	√	Onboard	Distributed	Maritime SAR	-	-
Centauro [9]	Locomotion and high power resilient manipulation platform	-	Onboard	Central	Harsh Environments	-	-
SmokeBot [23]	Robots with sensors for mapping in low visibility areas	√	Onboard	Central	Low Visiblity	-	-
TRADR [24]	Robots operate in SAR environments alongside humans	√	Onboard	Central	Industrial	-	-
UAV-UGV Construction [25]	UAV-UGV team that maps and plan around construction sites	√	Onboard	Central	Construction	-	-
Swarm Robotics Blockchain [20]	Simulation of Byzantine robots within robot swarm utilizing blockchain security	√	-	Distributed	Simulation	Blockchain	-
Wilderness SAR [26]	Simulation of wilderness SAR with proposed algorithm using Market-based Approach	√	-	Distributed	Wilderness (Simulation)	-	-
Our work	UAV maps the terrain and locates the target, cloud process the sensory images and sends the coordinates to the UGV	√	Offboard	Distributed	Simulation + Wilderness	Blockchain	√

**Table 2 sensors-22-05569-t002:** A succinct list of the features of IoT frameworks.

IoT Framework Company	Cryptography	Security Protocol
Samsung SmartThings [35]	128-bits AES protocol.	OAuth/ OAuth2 protocol.
AWS IoT Amazon [36]	128-bits AES + other primitive	X.509 Certificates + AWS IAM + AWS Cognito
Calvin Ericsson [37]	ECC protocol	X.509 Certificates + Sim-based Identity
Brillo/Weave Google [37]	Full disk encryption supported by Linux kernel	OAuth 2.0 + TEE
Kura Eclipse [37,38]	Multiple cryptography primitives	secure sockets
ARM Mbed [39]	mbed TLS + Hardware Crypto.	X.509 Certificates + other standards (mbed TLS)
HomeKit Apple [40]	256-bits AES + many others	Ed25519 public/private key signature + Curve25519 keys
Azure IoT Microsoft [37]	Multiple cryptography primitives	X.509 certificates + HMAC-SHA256 signature

**Table 3 sensors-22-05569-t003:** Network Opcodes.

Opcode	Data	Description
ADD_BLOCK	Public key of a new node	This message is sent when a new node is to be added to the chain. When nodes hear this message, they craft a new block with the public key that was in the message.
INTRUSION_ATTEMPT	IP address of illegitimate message sender	This is sent when a well-formed message is received that is not signed by a network member.
SENSOR_DATA	Any kind of data	This will carry important data received from the cameras and sensors connected to the rover.
BROADCAST_BLOCK	Public key of new node	This is used by the cloud computing node to send an add_block to each node.
ERROR	Description of error message	This is when a message is not formed correctly, and it lets the sender know something went wrong.
CONFIRMATION	Confirmation and the name of what is being confirmed	This is sent to let the sender know the operation was successful.

**Table 4 sensors-22-05569-t004:** Hyperparameters for the semantic segmentation model.

Key	Value	Description
Algorithm	FCN	The network used to create model.
Backbone	ResNet-50	The encoder used.
Crop Size	240	The image size for input during training.
Early Stopping	True	Regularization to avoid overfitting.
Early Stopping Min Epochs	20	The minimum number of epochs.
Early Stopping Patience	4	The number of epochs to meet before stopping.
Early Stopping Tolerance	0	
Epochs	50	The maximum number of epochs.
Gamma1 (decay factor)	0.9	γ factor used in (Equation 1)
Gamma2 (momentum factor)	0.9	α factor used in (Equation 2)
Learning Rate	0.000545237	η factor used in (Equation 2)
Learning Rate Scheduler	polynomial	The shape of the scheduler that controls its decrease over time
Number of Classes	4	The number of classes to segment, including the background.
Number of Training Samples	400	The number of images used for training.
Optimizer	RMSProp	The optimizer used to modify parameters.
Validation Mini Batch Size	32	The batch size fore validation.

**Table 5 sensors-22-05569-t005:** Comparison of the cloud model to efficient networks in previous works.

	ResNet-50 (Cloud)	LNSNet ^1^	ESPNet ^2^
GPU Utilization (%)	15	37	99
CPU Utilization (%)	30	11	21
Model Size (MB)	256	1.07	1.46

^1^ Performed on Intel i7-4710HQ and NVIDIA GeForce GTX 960M; ^2^ Performed on NVIDIA Jetson TX2.

## Data Availability

The dataset used in this case has been described in detail in the sections above. The researchers would be willing to provide more details if needed.

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
