# Peer review of "IoT Security and Computation Management on a Multi-Robot System for Rescue Operations Based on a Cloud Framework"

_sensors, 2022, doi:10.3390/s22155569_

Round 1

Reviewer 1 Report

1. Most of the literature referenced in this work is outdated and should be replaced with recent developments in the field from re-known relevant journals.

2. Although, the presented work is interesting and new, the authors failed to show the novelty of this in any of the sections.

3. Better to add a table to summarize the litterateur review carrying a comparison of security parameters improved in this work and different works in IoT security management. Also, add an overall system overview diagram in the introduction section.

4. Almost, all figures are poorly cropped and the text size in figures looks odd if it is larger than the normal text size. I suggest using some better tools to present figures. The text in Figure 3 is not readable and it's very challenging to understand what is happening in this figure.

5. Flow in the write-up is missing.

6. No clue given how the parameters given in Table 2 of semantic segmentation are set.

7. Figure 13 shows some text that is unclear to the reader.

8. Many grammatical issues and typos in the paper. Vague and unnecessary statements should be removed in the revised manuscript.

9. Results section does not have any comparison showing how, and what parameters are improved by the authors. Better to add a comparison table.

10. Almost, all sections need revision in terms of a technical research paper instead of a project dissertation.

Reviewer 2 Report

This is a very interesting topic regarding cloud computing for the computation management of unmanned ground vehicle (UGV) systems. The manuscript was easy to follow. However, several issues can be addressed, such as

-In the introduction, please provide a summary of the contribution and mention the major one as compared to the previous study.

-Instead of only citing the source, please elaborate more on each of the previous studies by adding a new section into the manuscript, such as Literature Studies. Contrast your proposed solution as compared to the existing one.

-The title was for IoT security, but there was no evaluation or design for this part. Please provide an assessment of how IoT security was developed in this study, as well as its performance.

-Figure 2 is too brief. How is blockchain used in this study? What kind of blockchain type did you use? 

- Please compare the performance of your system with and without the blockchain part, as you mentioned that the blockchain one is faster than the centralized one. Please prove it!

-Figure 3 is poor. Please use the appropriate font size and the quality of the image should be 300 DPI.

- There must be a delay time between the UGV and the cloud. Please provide the evaluation on this part since it is important to know the delay time between the data send from the UGV to the cloud, processing time in the cloud, and the final result given/returned back to the UGV.

- Most of the references are outdated. Please provide references in the last 5 years.

Round 2

Reviewer 1 Report

Maximum points are incorporated by the author. I suggest improvement in writup.

Reviewer 2 Report

There are no any concerns with the paper as they addressed all of the earlier comments.